# Digital Identity Levels in Older Learners: A New Focus for Sustainable Lifelong Education and Inclusion

José Manuel Muñoz-Rodríguez [1,*], María José Hernández-Serrano [1] and Carmen Tabernero [2,3,*]

1   Faculty of Education, University of Salamanca, 37008 Salamanca, Spain; mjhs@usal.es
2   Faculty of Social Sciences, University of Salamanca, 37007 Salamanca, Spain
3   Instituto de Neurociencias de Castilla y León (INCYL), Universidad de Salamanca, 37007 Salamanca, Spain
*   Correspondence: pepema@usal.es (J.M.M.-R.); carmen.tabernero@usal.es (C.T.)

**Abstract:** Identifying the digital identity of older adults entails an essential step for their effective digital inclusion grounded in a holistic and sustainable human development in hyperconnected societies. This paper proposes a theoretical framework with three levels of digital identity based on motives and practices: location, action, and significance. This framework was used for designing an ad-hoc scale, tested with a sample of senior learners ($n = 659$) aged 55 and over involved in active aging programs in Adult Education in Spain. Participants filled out a five-point 20-item Likert scale measuring their agreement with the digital identity factors, along with two complementary scales evaluating their internet uses and motivations, as well as other items on living arrangements and perceived social support. Exploratory and confirmatory analyses showed a factorial structure with three levels of digital identity for older adults. The results revealed that active older adults with diverse motives for using the internet and variety in digital practices recorded a higher digital identity level, as did those who felt more isolated or with less social support. As a conclusion, the educational implications according to instrumental, social, and motivational variables are key for the progressive construction of a digital identity in older adults and for their digital inclusion.

**Keywords:** digital technologies; older adults; digital identity; internet use; loneliness; active aging; digital inclusion

## 1. Introduction

The processes of older adults' digital inclusion are part of different international strategies within the principle of lifelong learning [1,2]. Moreover, the importance of education-for-all is among the priorities of the Incheon Declaration, adopted within the World Education Forum in 2015. A declaration [2] which represents a commitment to Goal 4 of the 2030 Agenda for Sustainable Development, spreading lifelong learning activities to all ages by creating "opportunities for equitable access to university for older adults, paying particular attention to vulnerable groups" (p. 41).

Furthermore, the governing principle enshrined in Agenda 2030 is that no one should be left behind; in other words, identifying the most vulnerable populations regarding the protection of their basic rights and needs. Today, social access to the internet is an issue that often leaves older adults behind, and acknowledging and analyzing their digital identity provide the necessary indicators for addressing the full social inclusion of older adults in the virtual arenas of co-existence and action, based on measures involving the digitization of their daily lives. The aim, according to the rationale behind Goal 11, is to consider the true social inclusion of vulnerable older adults' regarding access

to the internet, because their digital identity has not been recognized. Knowing older adults' digital identity means being able to draw up inclusive social action plans, considering the virtual environment to be a safe public arena, in line with Agenda 2030's target 11.7. The last Inter-American Convention on the Protection of the Human Rights of Older Persons precisely focused on the need to study this issue [3].

Under this premise, active aging has been described as an opportunity for economic interdependence and full participation in society [4,5]. Combining active aging with the emerging possibilities of learning technologies poses a challenge of unimaginable pedagogical and social scope via the attainable lifelong learning possibilities for adults and the elderly to continue learning throughout their lives [6].

Furthermore, education for development is the knowledge area that best argues in favor of this need. A society with more participative, active, and informed citizens is one of the core remits involved [7]. Knowing older adults' digital profile therefore helps us to lay the foundations for effective and sustainable education for development. Only by knowing an older person's digital identity can we consider their full inclusion in the digital society, whereby they have a critical and engaged view of reality; because it is within their reach, they know it. This full access will therefore enable us to guarantee a commitment and responsibility toward social action involving older adults and foster attitudes and values in local action through global citizenship. If facilitating an understanding of a global approach is one of the goals of education for development, this can only be achieved by accessing cyberculture, with prior knowledge of the digital identity of older adults [8].

A number of recent studies have focused on older adults' increasing access to different technologies [9–11], their engagement with digital technologies [3,12,13], or the improvements in wealth, health, and social relations due to the use of information and communication technology (ICT) [14,15], among others. The research has evidenced that even though stereotypes persist about older adults' lack of interest in the internet, and recognizing the digital cultural "lag" [16], more and more seniors are using the internet for a number of purposes, and thus take an active part in digital culture [17,18].

If advanced technologies are to provide sustainable learning opportunities for older adults and multiply their lifelong learning possibilities, there is a need to consider digital identity building and development, as this paper discusses. First, because even if identity is not an entity, it does manifest itself in different categories: forms, mechanisms, and processes, both complex and heterogeneous, associated with the temporal space coordinates in which subjects are situated during their life time, which should not be omitted during their practices and experiences in digital environments. Second, because the processes of identity-building depend on the nature of the actions undertaken or experienced vicariously or not, or even imagined, in which there are motives. Such is the case of the motivations for using specific technologies or online browsing. Hence, the term "identification practices" is sometimes preferred to the term "identity," as these practices lend meaning to the subject's behavior and life [19], and thus the process of being is determined by how and where the subject transitions between, and lives in, spaces and, therefore, also in digital environments [20].

The question remains as to why online learning possibilities are infinite for younger generations while appearing to be limited for older adults. Rather than a difference in online skills, practices for identification make the crucial differences. This paper therefore argues that, more than a digital divide or "grey divide" [21], there is an identity divide which determines the "digital being" of learners of different ages.

Identity is an attractive and extensively studied concept that is sufficiently grounded in theoretical and applied outcomes in the research into social and human scientific fields [22]. While some disciplines do not consider the concept of identity to be an adequate analytical category [23], for others it is an integral part of the study of human experience [24] or the core remit of lifelong educational processes [25]. The constructivist approach is widely accepted because identity is a process that constantly reconfigures the self and is also socially constructed by others. It is not an attribute, but instead a relational process. There is, therefore, no single identity, but multiple and varied identities,

changes, and dispositions through which subjects act and react. Nevertheless, more focus needs to be given to ways in which the construction and development of identity is carried out within online environments, especially for older adults, who research has already viewed in terms of an "aging identity" [26] or an "active identity" [27], but not a digital identity. Identity is a polyhedral concept that directly affects the person and their social projection, both material and symbolic [28].

The originality and significance of this research is found in the older adults' inclusion as potential digital learners, considered not as a homogeneous generation [4,26,29], but as a group with diverse digital profiles and motivations, and considering them not as mere users, but as learners undergoing a different process of digital identity construction, adjusted on at least three levels, and grounded in a sustainable, holistic, and inclusive development. The main goal of this research is to evaluate the role played by the digital identity levels considering the relationship with the motivations that older learners have when going online, along with other instrumental or social variables. Accordingly, these are the main research questions:

1. RQ1: Are instrumental variables regarding the use of technologies (accumulative time in using the internet and frequency of connection) associated to the lower or higher level of digital identity in older adults?
2. RQ2: Are social variables (living arrangements, perceived social support, and loneliness) associated to the lower or higher level of digital identity in older adults?
3. RQ3: Are motivational variables (connecting purposes or motives to use the internet) associated to the lower or higher level of digital identity in older adults?

A theoretical framework that supports and justifies the (re)construction of older adults' digital identities was developed for this purpose and to answer these questions, then tested for validation via an ad-hoc questionnaire with a sample of senior learners involved in active aging programs. The involvement (correlation) of instrumental, social, or motivation variables would prompt suggestions towards a sustainable adult education to better accommodate the levels of digital identity and motivations that older learners have for going online for their effective digital inclusion.

## 1.1. Three Levels of Digital Identity in Older Adults: A Theoretical Framework

The internet and social media are spaces of practices for identity construction, described by Turkle [30] as a "social laboratory, where people experiment with constructions and reconstructions of self" (p. 27). Nonetheless, the possibilities of a digital identity are more than a transposition, a reference to avatars, and a virtual representation of a self [31,32]. From a psycho-pedagogical perspective, digital identity development refers to individual and social processes of mediation and negotiation with oneself, with others, and with digital culture.

Because of the generation gap, it is not surprising that digital identity has been more widely studied in younger generations [33] and that little research has focused on digital identity in older adults [20]. Our proposal is based on the models of identity development in adulthood [34,35], with four logical assumptions: (1) identity is constructed and affects all humans; (2) identity is something that everyone has or should have as a social imperative; (3) identity is not always consciously constructed; (4) identity implies leaving marks and traces that differentiate people from each other, both individually and collectively. In our framework, these assumptions are transposed to the digital setting in order to avoid a reification of identity, and with the aim of developing a comprehensive understanding of the digital identity processes undertaken by older adults. Instead of a unique identity, older adults' interaction with the digital world enables them to construct changeable identities, not always consciously, outside of or self-determined within the digital culture, and with differential traces and scripts.

The proposal here involves a continuum of three levels (see Figure 1) based on several models of identity, allowing us to comprehend a great deal of older adults' ways of being and seeing, of participating and relating, and of feeling and creating in digital scenarios. On the one hand, our framework is related to the multidimensional taxonomy of identity that also recognizes three levels: individual subjectivity

(self-identity), an individual's specific behavioral patterns (personal identity), and membership of social groups (social identity) [25,36]. On the other hand, our framework is also connected with the model of transits in the construction of the identity state [37], along with different traditional theories that have justified models of personal development and identity. These include the theory of the anthropological place [38], the theory of social action and ubiquity [39], symbolic interactionism [40], the theory of communicative action [41], and the social construction of reality [42]. Our model also brings together different notions about identity, such as those related to the classic identity [43], the paradigm of identity status [34], or the social development of identity [24], among others.

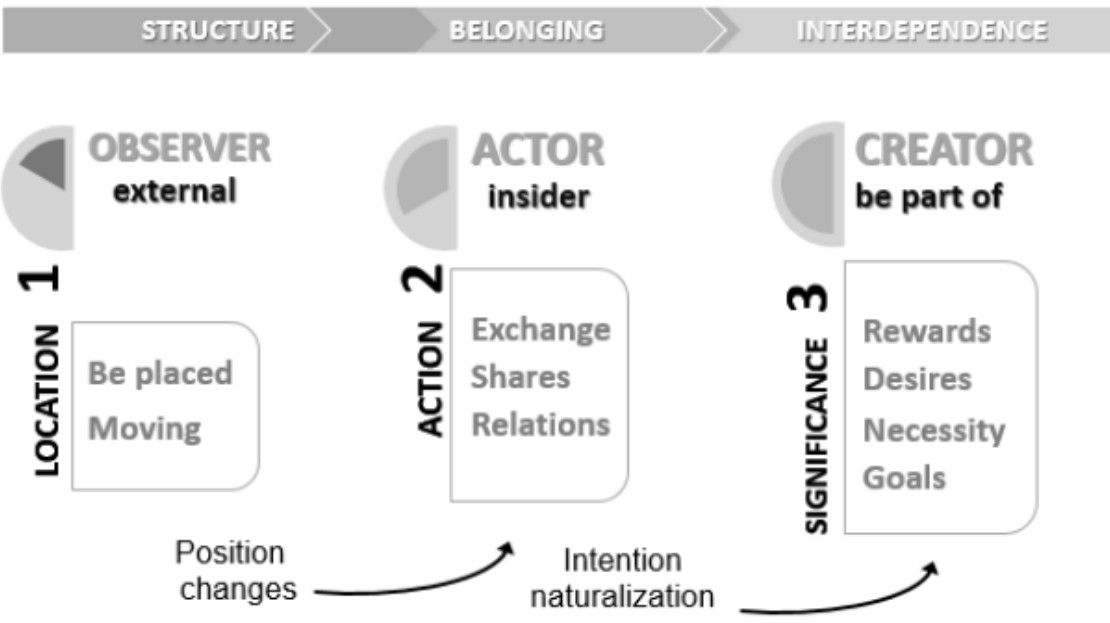

**Figure 1.** Theoretical framework: three levels of digital identity.

As depicted in Figure 1, there are three levels of identity (location, action, and significance) through which the older adult can progress or withdraw in terms of digital practices and experiences across multiple environments and purposes. The first level is for those acting as observers of the digital culture, external to or ill-identified with the environment, but aware of a structure or of possible movements. Through in–out position changes, level two is achieved, whereby they become autonomous players inside the virtual environment, belonging to it, and recognizing others with whom to interact and relate. Finally, when naturalization of the environment is reached, in level three, players become the creators of their own digital identities, and interdependently part of the environment, with intentions and rewards for motivation, engagement, and learning. Levels are theoretically explained in-depth below.

### 1.1.1. First Level: Location Identity

This level represents the stage at which older adults' actions are fundamentally focused on entering or leaving a virtual environment. It constitutes a process of orientation as a basic behavior of being positioned unidirectionally, without any further in-depth actions. It is linked to the first logical assumption of Sica, Aleni, Syed, and McLean [35], as described, because we are spatial–temporal beings, and any human process relies on location by means of a sense of space and time, in which any action could be comprehended [44], and from which our actions are tracked to give us reasons for what has been done and what remains, in line with the fourth logical assumption.

It is essential for older adults to be digitally located in order not to feel out of place, linked with Augé's theory of anthropological space [38], because the occupation of space–time coordinates allows them to position themselves. However, due to access barriers, a previous level could exist—a zeroth level of identity—on which older adults are frustrated in this initial step of location, and are unable

to have a sense of being in the environment, because the very action of entering or leaving poses a challenge. Thus, there is no possibility of familiarity, rewards, or any sense of identification without the understanding of location with possibilities of movement [37]. On this first level, digital culture may constitute something external to older adults' life, sometimes uncertain, at other times even intimidating [45]. Older adults may understand diachronic events of digital culture externally, making the digital world appear to be an impersonal, unstable place, even broad or extensive, and sometimes unstructured. However, external experience when they are located will enable them to structure their personal and social biography. This "being placed" is the correlate of "to move", leading to new forms of cultural expression and identity-building, as stated by the rational materialist approach [46].

In other words, external experience implies an intelligibility for the things that are, that can be done, or that others can do. It is a know-how that lets older adults experience satisfying feelings or rewards, allowing them to approach the next level. Location is, therefore, the foundation of digital identity for the elderly. Nevertheless, for some older adults, this first level could also be a permanent state, following Marcia's identity status paradigm [34], if they put limits to their being on the internet [47], with no room for adaptive processes; as a consequence of such accidental or unstable location, older adults would move like passengers or explorers towards the next level, where others with whom to interact are perceived. The older adult needs a "position change", which would bring further possibilities of participating, relating, and doing with a greater involvement in the digital environment.

### 1.1.2. Second Level: Action Identity

This second level represents actions within a digital environment, along with techno-mediated communications with others, following the approaches of Côte [25] and Schwartz, Luyckx, and Vignoles [36]. The digital identity explanation focuses not on the older adult's mindset in the face of digitality (dualist epistemology), but on social relationship patterns when participating in the digital environment (social epistemology) [48]. Based on the theory of social identity [49] and the social construction of reality [42], this level implies managing the social self that involves a point of encounter between the individualities and the influence of a community that regulates the idiosyncrasy of their life stories according to social demands. It is the beginning of a social representation, as a "social self" exposed to the perception of others [50]. According to the theory of social action [39], there is also a need to manage or establish an "order of interaction," look for "social encounters," and make shared decisions for "effective cooperation", all of which create a sensation of the presence of others and reciprocal influence, lending relevance to the concept of a "social situation."

In this level, the external aggregation, artificiality, and depersonalization evident in the first level give way to proximity, the possibility of sharing actions, and interacting with others. Nevertheless, the closeness to others can meet functionalist aims (superficial social identity) based on external purposes and automatic responses, or symbolic interactionist aims (complex social identity) based on inner motivation and subjective constructions about the relationship between oneself and others [51]. Based on the latter, digital identity will be the result of socially accepted and individually internalized relationships and exchanges, from Jenkins' social identity development theory [24].

The sense of reciprocal and joint actions informs a link which elicits and informs the sense of social identity [52]. Older adults' digital identity is created and transformed through relationships with other people and with the social representations created by others [53]. This sense of social feeling allows them to make and share things, generating the experience of community. From the feeling of being accompanied, each exchange inevitably generates a reference index to the others [54], so social influence and reciprocity can be expected at this level of digital identity construction.

Furthermore, connection mediated by digitality allows for communicational acts [55], understanding communication fully as discursive practices, not only linguistics, as Habermas [41] has stated in his communicative action theory. The idea of the impossibility of noncommunication [56] is reproduced in the digital environments and leads to "participatory communication" and Mead's symbolic interaction

theory [40], supported mainly in the expressive function in which older adults can participate in multiple messages through situational diversity. Again, this communication, participation, and social sharing could be understood by older adults as an end process for constructing their digital identity, stopping at this second level of action. Alternately, they could be helped to develop a higher level, in which involvement within the digital environment attains higher significance, because choices are intentional and interactions are lived as natural.

### 1.1.3. Third Level: Significance Identity

This level is the result of a progressive process, as a meditated configuration that provides meaning for themselves and their actions [57]. Digital experiences rediscover the necessary chronotopic referent for older adults to be fully developed and integrated into digital culture. A state of digital self-realization could be reached, even self-determination, because they are naturalizing digital actions as significant to their lives. As a result of these identification practices [19], digital identity is effective because of what Erikson [58] called "synthetic capacities of the self," when older adults incorporate and endorse the capital of digital culture. This level contains the so-called intangible identity resources that include meanings, memories, sensations, and cognitions generated while participating and relating in digital environments, whose subjective significance will develop into their own style of digital identity [59]. This is a subjective level of identity, regarding the third logical assumption of Sica, Aleni, Syed, and McLean [35] and Marcia [34], which arises from perceptions that each older adult can translate into a different feeling of belonging with a desire for permanence, or the need for connection.

At this level, location has evolved into a virtual geographical entity, using gerontechnology terms [60], whereby older adults could designate the meaningful virtual places in their life with specific names. Older adults build discourses, scripts, and memories that imbue them with affectivity and rational meanings. The digital identity of older adults is also built on the basis of a temporary self-management, following the classical models of Berzonsky [43] and Jenkins [24], namely, historicity begins to exist by establishing memories and creating a sense of time. There is synchrony in addition to diachrony, because older adults can project the dialectical process as diachronic traces loaded with emotions, feelings, and preferred or avoided places [54]. They have a history of frequented sites where they would like to stay or return, towards the anachronistic sense of being comfortable; this can only happen when they have a sense of inhabiting a social space for their development, as indicated by Bullingham and Vasconcellos [31] in their theory of virtual representation.

Older adults at this level have learned to decipher the hypertextual polymorphism produced by the mixture of meanings, objects, actions, and people that intersect on the web [61]. Being online in different digital environments becomes such a significant process that they are what semiotic theory calls "interpreters" of themselves and others on the internet [62]. They adjust to the environmental circumstances not only autonomously, but also creatively, by acquiring natural habits immersed in meaningful schemes that guarantee significance [63]. They develop not only a semantic and logical understanding of form and content, but also of their symbolic and metaphorical meanings. There is a continuity in the identity development between virtual and non-virtual environments, because digitality can be seen as an interdependent part of the whole process of identity reconstruction.

## 2. Materials and Methods

### 2.1. Participants

University programs for adults and people aged 55 and over in Spain are the initial threshold for referring to active aging [64–66]. Since there are very low-cost non-formal education courses, teaching by university academics, with wide and updated syllabuses, which do not award any of the official qualifications in the formal system but instead a certificate of attendance. Accordingly, these students were chosen for the study precisely because they are either still working or involved in other unpaid social activities, leading a healthy, independent, and confident life [67,68]. In addition, they are active

older adults with the ability to participate and learn, being independent and confident, with the enough internet experience and uses to complete the scale and test the research questions on correlations.

A total of 659 older adults aged 55 and over participated in this study via incidental convenience non-probability sampling (see Table 1). All the participants were enrolled in formal lifelong learning programs at nine of Spain's adult universities. The gender distribution of the sample was balanced (50.7% male, 49.3% female). The participants fell into different age ranges (11.5% under 60, 34.6% between 61 and 65, 34.7% between 66 and 70 years, 14.9% between 71 and 75, 2.7% between 76 and 80, and 1.5% over 81). As regards residential status, 75.6% lived in towns or cities of more than 50,000 people. There were no statistically significant differences by gender or age according to the adult university distribution.

**Table 1.** Sample's demographic characteristics and internet access patterns.

| *n* = 659 | | *n* (%) |
|---|---|---|
| Gender | Female | 325 (49.3) |
| | Male | 334 (50.7) |
| | <60 years | 76 (11.5) |
| | 61–65 years | 228 (34.6) |
| Age | 66–70 years | 229 (34.7) |
| | 71–75 years | 98 (14.9) |
| | 76–80 years | 18 (2.7) |
| | >81 years | 10 (1.5) |
| Age of the Internet Usage Initiation | <5 years | 50 (7.6) |
| | 5–10 years | 97 (14.7) |
| | >10 years | 512 (77.7) |
| | 1 or 2 days | 55 (8.3) |
| Daily Internet Usage | 3 to 5 days | 110 (16.7) |
| | Only weekends | 6 (0.9) |
| | everyday | 488 (74.1) |
| Using Internet on their job, Previously or Currently | Yes | 545 (82.7) |
| | No | 114 (17.3) |

In relation to living arrangements, 23.4% of participants lived alone, 64.9% were living with a partner, and the remaining participants were living with others (7.4% with a son or daughter, 3.2% with another family member, 1.1% with other people). As for employment status, 80.4% of participants were retired or pensioner, 6.1% were employed, 6.1% were unemployed, and 7.4% were in another, unspecified labor situation. When asked whether they had to use the internet when they were employed, 82.7% of participants affirmed that they had. Regarding participants' level of education, 58.4% had completed a bachelor, master's, or doctoral degree; 33.2% had a secondary education degree; 8.2% participants had primary education; and one participant (0.2%) had no formal education.

## 2.2. Procedure

After the program coordinators at nine universities were informed about the aim of the study, an invitation to participate was submitted by email to all senior students from university coordinators. The invitation message and questionnaire were developed in Google Docs with a link that included the following instruction: "This questionnaire aims to measure students' perception about their digital identity (Who am I on the internet) when they connect, identify, and/or interact in different services and networks. The answers will be treated confidentially. Answering it will not take more than 10 min." The questionnaire included several questions about socio-demographic variables and three scales to evaluate digital identity, as well as a validated scale of motives for using the internet and 12 questions on their browsing habits. Answers were collected over the span of a month.

Although surveys using online formats [69] have some strengths (e.g., flexibility, speed and timeliness, ease of data entry and analysis), it is also important to highlight certain weaknesses of using this methodology (e.g., impersonal context, privacy, and ethical issues). In this sense, according to the general data protection legislation, this study sought to ensure the anonymity, privacy, and confidentiality of answers. Before starting, participants were presented with brief instructions requesting their consent and informing them that they could leave the survey at any time. No personal data were requested.

*2.3. Measures*

In order to discriminate participants by segment, two sociodemographic questions about gender and age were asked. Then, based on the Multidimensional Scale of Perceived Social Support [70], two items were selected to assess the perceived social support and loneliness in their context ("I feel that I have people to talk to and discuss my concerns"; "If I need help, I consider that I have people to go to"). The participants responded using a five-point Likert scale (1 = never; 5 = always). In addition, based on the UCLA Loneliness Scale [71] the questionnaire asked about the level of feeling of loneliness ("To what extent do I feel alone?") and participants responded using the same five-point Likert scale. The three items recorded good reliability ($\alpha = 0.71$); therefore, after reversing the item of loneliness, a single variable was created that we call "social support", according to which the higher the score, the more social support and less loneliness perceived by the participants.

Two questions referring to the accumulation of time were incorporated to evaluate participants' level of internet use [72]: the length of time they had been using the internet (less than five years, 5–10 years, more than 10 years) and the number of days per week they connected to the internet (one or two, three to five, only on weekends, or every day). Both items showed a high correlation ($r = 0.64$, $p < 0.001$), so a combined measure was created according to which the higher the score, the more time spent online.

To evaluate the reason each participant connects to the internet, 12 items were generated to record the frequency of different activities (e.g., "frequency with which you are connected to find information") based on the study by Llorente-Barroso [73] for the same population. Participants answered using a four-point Likert scale (1 = never, 2 = rarely, 3 = frequently, 4 = continuously). An exploratory factor analysis with Varimax rotation showed that three principal components explained 59.21% of variance. The first factor, transactional purpose, is based on the digital actions of banking, managing, buying, and entertainment (39.69% variance, reliability index $\alpha = 0.78$). The second factor, active participation purpose, is based on the digital actions of information sharing, giving opinions, keeping updated, and training (11.10% variance, reliability index $\alpha = 0.72$). The third factor, informational purpose, is based on searching for information, receiving communications by email, and reading newspapers online (8.42% variance, reliability index $\alpha = 0.70$).

The Internet Motivation Scale developed by Wolfradt and Doll [74] was used to evaluate older adults' main motivations for going online. The scale consists of 20 items that evaluate the three fundamental reasons that people have for connecting to the internet: motivation for information (eight items with a good reliability, $\alpha = 0.87$, e.g., "I consider the internet as an additional mass medium"), motivation for interpersonal communication (seven items with good reliability, $\alpha = 0.86$, e.g., "I use the internet to express myself"), and motivation for entertainment (five items with good reliability, $\alpha = 0.84$, e.g., "The internet stimulates my curiosity"). In the original scale published by Wolfradt and Doll [71], the reliability indices were similar to those found in our sample (0.84, 0.81, and 0.76, respectively). Participants had to respond to the scale using a five-point Likert scale, where 1 = completely disagree and 5 = completely agree.

Finally, the Digital Identity Questionnaire for Older Adults was created to analyze participants' digital identity according to the theoretical framework proposed. To create this instrument, an initial list of 20 items was produced, based on the main identity dimensions theoretically discussed for the three levels. Content validity was tested by a Delphi panel consensus with ten experts in the fields

of education, psychology, and technology, who agreed on content and comprehensibility validity by adapting a grounded theory with a reformulation of items for methodological and conceptual precision. The final questionnaire was structured as follows:

1.  Level 1: Location identity (eight items measuring convenience, location, familiarity)
2.  Level 2: Action identity (seven items measuring friendship, membership, sharing, socializing, and loneliness).
3.  Level 3: Significance identity (five items measuring bonds, interdependence, necessity, and completeness).

Participants had to respond to the questions using a five-point Likert scale where 1 = completely disagree and 5 = completely agree. Piloting tests were previously held by observing doubts and suggestions in a face-to-face meeting with 30 senior learners to improve the scale for validity and readability.

## 2.4. Statistical Analyses

Before analyzing the factors that influence the motivations of older learners when connecting to the internet and evaluating the role played by the digital identity of older adults, a digital identity questionnaire was created, and its psychometric properties were tested. To cover this, an EFA and a CFA were conducted with SPSS v25 and Amos v25. First, with a view to reducing and refining the digital identity scale, all items were subjected to preliminary data checks to explore their suitability for inclusion in further analysis. This was followed by a reliability analysis to identify items that considerably decreased the internal consistency (Cronbach's alpha value).

To validate this new measure, an exploratory factor analysis (EFA) was performed, followed by a confirmatory factor analysis (CFA). We carried out an EFA with Varimax rotation to identify subscales within the item pools and to exclude items that did not group in conceptually sound subscales. The suitability of using factor analysis was assessed using Bartlett's test of sphericity (BTS) and the Kaiser–Meyer–Olkin (KMO) statistic. A KMO value of 0.50 or higher is considered acceptable for a satisfactory factor analysis. For the BTS, a *p* value of 0.05 or lower serves as the criterion for indicating that a factor analysis can be implemented. We subsequently performed a CFA using Amos 25. The following indices were used to test how well the model fits the data: chi-squared ($\chi^2$); the minimum discrepancy, divided by its degrees of freedom (CMIN/DF); the normed fit index (NFI), the comparative fit index (CFI); the root-mean-square error of approximation (RMSEA); and the Tucker–Lewis index (TLI). Internationally accepted rules were considered for interpreting the goodness of fit of the different indices. The correlations between digital identity (and its subscales) and other motivational and psychosocial variables were investigated to explore the external validity of the final scale.

Chi-squared tests were used to evaluate the relationship between the digital identity dimension and the sociodemographic variables (RQ1), Correlational analyses were performed to evaluate the relationship with continuous variables (motives, purposes, and loneliness). Different ANOVAs were performed for RQ2, to evaluate the differences between those who were living alone or with others on perceived social support, and the differences between those who used the internet in their job or not on digital identity. Finally, for RQ3, correlation analyses were run to evaluate the relationships in connecting purposes and motives to use the internet with the levels of digital identity in older adults.

## 3. Results

### 3.1. Exploratory and Confirmatory Factor Analysis

Initially, 20 items were designed to evaluate the level of digital identity of active older adults; first, an EFA was performed with principal component methodology and a Varimax rotation with Kaiser normalization. Six items were removed to improve the reliability of the factors and fit the criterion that the loadings of all the items were higher than 0.40. The sampling adequacy of the data

for factor analysis indicates that the Bartlett test for the contrast of sphericity allows us to reject the hypothesis that identity is the population correlation matrix (BTS ($\chi^2$ = 6221.39; d$f$ = 91; $p < 0.001$). The value of the Kaiser–Meyer–Olkin test (KMO = 0.94) suggests that the correlation matrix is adequate to continue with the factor analysis. Three principal factors were obtained, explained by a 71.73% variance (see Table 2). The first factor corresponded with the action identity level and was explained by five items, which showed an adequate reliability ($\alpha = 0.89$). The second factor corresponded with the localization identity level and was explained by five items that also showed an adequate reliability ($\alpha = 0.89$). The third factor corresponded with the significance identity level and was explained by four items, again showing an adequate reliability ($\alpha = 0.88$).

**Table 2.** Mean and standard deviation of digital identity questionnaire items resulting from the EFA extracting three factors.

| Items | Factor | | | Mean | SD |
|---|---|---|---|---|---|
| | 1 | 2 | 3 | | |
| Item 2 [Being on the Internet is a Rewarding Process] | 0.239 | **0.813** | 0.282 | 2.80 | 1.10 |
| Item 1 [The Internet Seems to Me a Cozy Place] | 0.224 | **0.808** | 0.209 | 2.66 | 1.08 |
| Item 3 [I Enter, I Go Out, and I Am on the Internet as if I Were "at Home" | 0.152 | **0.808** | 0.235 | 2.84 | 1.25 |
| Item 5 [on the Internet, There Are Already Many Familiar Sites, with which I Feel Identified] | 0.452 | **0.582** | 0.188 | 2.58 | 1.17 |
| Item 7 [When I am on the Internet, I Have the Feeling of "Being There"] | 0.494 | **0.536** | 0.355 | 2.17 | 1.11 |
| Item 12 [The Internet Allows Me to Have or Expand Relationships/Friendships] | **0.848** | 0.169 | 0.176 | 2.21 | 1.15 |
| Item 14 [When I Am Connected, I Have the Feeling of Being Part of a Group] | **0.815** | 0.216 | 0.282 | 2.17 | 1.11 |
| Item 13 [The Internet Makes it Easier for Me to Exchange Experiences, Ideas and Emotions] | **0.788** | 0.245 | 0.267 | 2.52 | 1.23 |
| Item 15 [I Like to Share Information on the Internet] | **0.728** | 0.200 | 0.175 | 2.24 | 1.21 |
| Item 11 [When I Am on the Internet I Do Not Feel Alone] | **0.661** | 0.311 | 0.229 | 2.18 | 1.09 |
| Item 18 [Internet Connection is a Necessity] | 0.187 | 0.294 | **0.838** | 2.26 | 1.26 |
| Item 20 [I Feel Linked to the Internet] | 0.359 | 0.375 | **0.742** | 2.27 | 1.21 |
| Item 19 [I Find Calm Being Connected] | 0.415 | 0.268 | **0.718** | 2.02 | 1.06 |
| Item 16 [I Want to Spend Time Connected to the Internet] | 0.282 | 0.571 | **0.462** | 2.72 | 1.10 |
| % Explained Variance | 28.33 | 24.92 | 18.48 | | |
| Reliability Index | $\alpha = 0.89$ | $\alpha = 0.89$ | $\alpha = 0.88$ | | |

Second, a CFA was performed to verify the factorial structure in three factors (see Figure 2). The model showed a satisfactory fit index [$\chi^2$ (d$f$ = 53) = 155.021; $p < 0.001$; CMIN = 2.925; NFI = 0.976; IFI = 0.984; TLI = 0.968; CFI = 0.984; RMSEA = 0.054 (LO 90 = 0.044/HI 90 = 0.064)].

In sum, the psychometric properties of the scale proposed showed three principal factors, where a first factor corresponded with the action identity level, a second factor corresponded with the localization identity level, and a third factor corresponded with the significance identity level (see the Supplementary Material, Annex with the validated Scale).

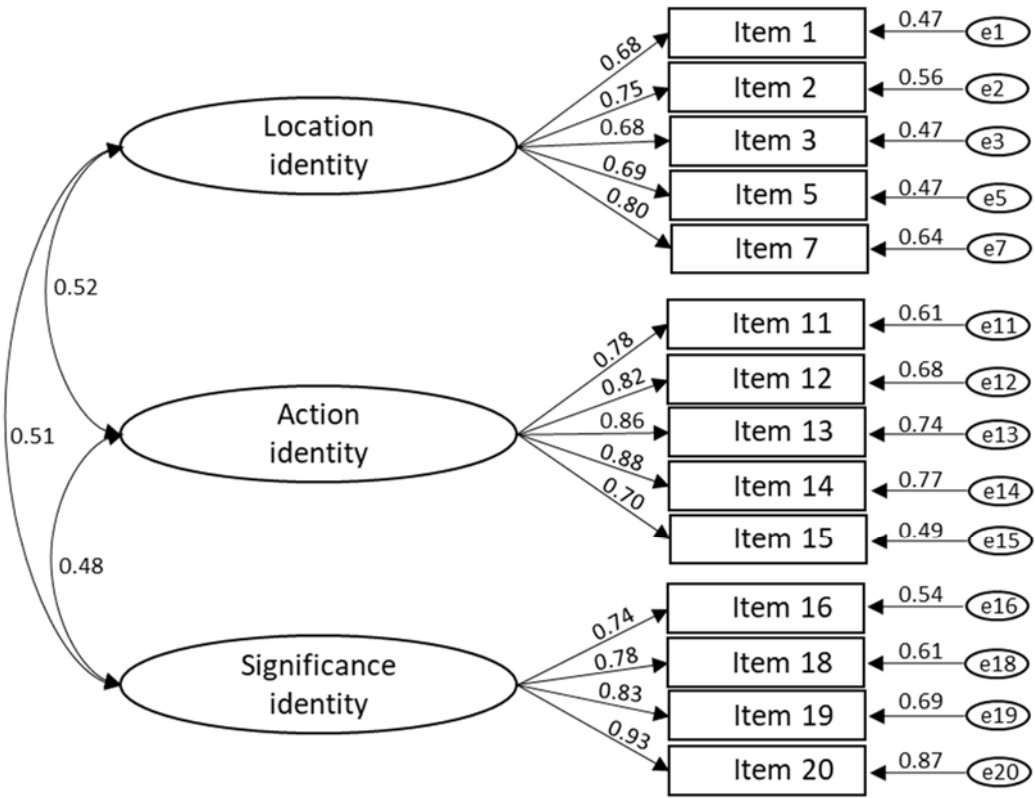

**Figure 2.** CFA or digital identity according to the three-factor dimensional structure.

### 3.2. Differences in Instrumental Variables on Digital Identity Levels (RQ1)

The correlation between the three factors of digital identity and the use of the internet, based on the amount of time the participants have been online and the number of days per week they connect to it, showed a statistically significant relationship with levels of location ($r = 0.23$, $p < 0.001$), action ($r = 0.10$, $p < 0.02$), and significance ($r = 0.23$, $p < 0.001$). Those participants who reported using the internet for work in the past showed a digital identity associated with the location level ($F(1,657) = 5.13$, $p < 0.05$, $\eta^2 = 0.01$; $M_{no} = 2.43$, $SD = 0.94$; $M_{yes} = 2.65$, $SD = 0.94$) and the significance level ($F(1,657) = 7.00$, $p < 0.01$, $\eta^2 = 0.01$; $M_{no} = 2.10$, $SD = 0.98$; $M_{yes} = 2.37$, $SD = 0.97$). The latter also use the internet more frequently than those who had not used it for their job ($F (1,657) = 151.92$, $p < 0.001$, $\eta^2 = 0.19$; $M_{no} = 2.53$, $SD = 0.94$; $M_{yes} = 3.26$, $SD = 0.47$).

### 3.3. Differences in Social Variables on Digital Identity Levels (RQ2)

A preliminary analysis to evaluate the relationships between living arrangements and perceived social support showed that participants living alone reported having fewer people nearby with whom they could discuss concerns ($F(2,656) = 10.86$, $p < 0.001$, $\eta^2 = 0.03$; $M_{alone} = 4.08$, $SD = 0.96$) or ask for help ($F(2,656) = 7.09$, $p < 0.001$, $\eta^2 = 0.03$; $M_{alone} = 4.31$, $SD = 0.92$). They also felt more isolated ($F(2,656) = 17.27$, $p < 0.001$, $\eta^2 = 0.02$; $M_{alone} = 3.71$, $SD = 0.96$) than those who lived with their partners ($M_{partner} = 4.47$, $SD = 0.84$; $M_{partner} = 4.59$, $SD = 0.75$; $M_{partner} = 4.17$, $SD = 0.81$, respectively) or with other people ($M_{others} = 4.39$, $SD = 0.89$; $M_{others} = 4.53$, $SD = 0.79$; and $M_{others} = 4.07$, $SD = 0.92$, respectively). The Scheffe post hoc tests showed statistically significant differences in the three questions between those who lived alone and those who lived with their partners. The differences remained significant with the global measure of social support ($F(2,656) = 18.49$, $p < 0.001$, $\eta^2 = 0.05$; $M_{alone} = 4.03$, $SD = 0.74$; $M_{partner} = 4.41$, $SD = 0.62$; $M_{others} = 4.33$, $SD = 0.71$). The differences between those who lived with partners and those who lived with others were not statistically significant at the level of people nearby to talk about concerns.

Correlational analysis showed that those participants who felt more isolated and had fewer people close to them showed a higher digital identity related to the significance level (r = 0.09, $p < 0.05$, and r = −0.09, $p < 0.05$, respectively); this relationship was statistically significant with the variable created to evaluate perceived social support (r = −0.09, $p < 0.05$). However, this relationship was not statistically significant with the other levels of digital identity, location (r = 0.01, *n.s.*), and action (r = −0.02, *n.s.*).

### 3.4. Differences in Motivational Variables on Digital Identity Levels (RQ3)

Correlational analyses were performed among all the motivational variables related to internet use: motives for using the internet, digital identity levels, and reasons for going online, finding higher positive correlations (see Table 3).

**Table 3.** Means, standard deviations, and correlation index of motivational variables studied (motivations, purposes, social support) and Digital Identity Levels (location, action, significance).

| Variables | 1 | 2 | 3 | 4 | 5 | 6 | 7 | 8 | 9 | 10 |
|---|---|---|---|---|---|---|---|---|---|---|
| 1.- Motiv. Information | 1 | | | | | | | | | |
| 2.- Motiv. Interpersonal Communication | 0.48 ** | 1 | | | | | | | | |
| 3.- Motiv. Entertainment | 0.55 ** | 0.70 ** | 1 | | | | | | | |
| 4.- Dig. Ident. 1 Location | 0.54 ** | 0.69 ** | 0.72 ** | 1 | | | | | | |
| 5.- Dig. Ident. 2 Action | 0.38 ** | 0.70 ** | 0.57 ** | 0.65 ** | 1 | | | | | |
| 6.- Dig. Ident. 3 Signific. | 0.45 ** | 0.64 ** | 0.69 ** | 0.76 ** | 0.66 ** | 1 | | | | |
| 7.- Purpose Information | 0.55 ** | 0.27 ** | 0.33 ** | 0.33 ** | 0.21 ** | 0.28 ** | 1 | | | |
| 8.- Purpose Transactions | 0.41 ** | 0.28 ** | 0.31 ** | 0.28 ** | 0.18 ** | 0.26 ** | 0.60 ** | 1 | | |
| 9.- Purpose Active Participation | 0.39 ** | 0.44 ** | 0.33 ** | 0.35 ** | 0.43 ** | 0.32 ** | 0.45 ** | 0.48 ** | 1 | |
| 10.- Social Support and Loneliness | 0.09 * | −0.06 | −0.09* | 0.01 | −0.02 | −0.09 * | −0.13 ** | −0.09 * | −0.09 * | 1 |
| **M** | 3.60 | 2.24 | 2.82 | 2.61 | 2.27 | 2.32 | 3.18 | 2.17 | 2.01 | 4.31 |
| **SD** | 0.74 | 0.83 | 0.89 | 0.94 | 0.97 | 0.98 | 0.61 | 0.63 | 0.70 | 0.68 |

\* Correspond to $p < 0.05$; \*\* correspond to $p < 0.01$.

The relationships revealed the expected importance of diversity in online practices; those older adults with more motives achieved a higher digital identity level and online use for participatory purposes; in other words, motivation reasons (information, communication, and entertainment) led to a higher digital identity level and also reflected online use not only for information or transactions but also for active participatory purposes. However, the correlational analyses also showed that those who perceived more social support and less loneliness were more motivated to use the internet for information, while participants who perceived low social support and higher loneliness reported motives for entertainment, the highest digital identity level, and connecting purposes.

## 4. Discussion and Conclusions

Evidence of the digital identity development in levels in active older adults provides an innovative focus, which can be useful for adult educators, teachers, virtual learning environment designers, among others, designed for the sustainable digital inclusion of older lifelong learners.

The results showed that those older-adult learners who recorded more internet uses and more diversity in motivation reasons for going online, achieved a higher digital identity level, together with those who perceived higher isolation; this diversity in uses could drive them to the "identification practices" [19], which help them by compensating for the lack of face-to-face social support. Thus, Adult Education programs that evolve or need to take advantage of the online learning possibilities should consider the digital identity dimension, which has rarely been considered before for older adults [75] and not previously related to social or motivational variables for the use of online technologies, as this paper studied. Furthermore, the possibilities of using the virtual environments for delivering learning in Adult Education should consider the social and motivational variables,

which will favor not only the progressive digital identity in older adults but also their sustainable development and digital inclusion.

The findings here reveal that the higher the level of digital identity, the greater the possibilities of social actions and, therefore, of social mobilization, one of the four main dimensions of education for development. A higher level of digital identity is the expression of older adults' greater active involvement [3]. This leads to education's direct role in the promotion of sustainable human development, thereby adding education for development as a mainstay of active aging. From this perspective, we are laying the foundations for dimensioning education for development as a way of facilitating older adults' more pro-active engagement as citizens in favor of social transformation [76].

The results support other studies that reveal the parallelism and links between active aging and sustainable development [77–79]. There is a clear need to relate social education and adult education to sustainable human development, prompting educational actions tied to equity and equality, democracy, and a cohesive society. Stress is placed on this sense of belonging, whereby local communities, including older adults, embrace the leading role corresponding to them in their human development processes to ensure they can effectively play an active role in social processes, including those that take place online. The complexity of virtual environments for older adults leads from the model of sustainable development to a global paradigm of citizenship that emphasizes these adult's necessary social commitment on the internet [64,80].

Our research supports other studies [81,82] on the relevance of social variables, as online sociability could reduce loneliness in older adults and marshal social support in online social networks [9,83]. This is a key point for the design of online environments for older adults that needs to focus on social encounters and rewards as constructive elements in older adults' digital identity. In our study, the older adults who perceived higher isolation reached the highest level (significance); accordingly, the design of online environments for them should be a catalyst for collective interactions to feel linked to others, participating in shared relational and sociable schemes, via activities that promote negotiations, creativity, and social understanding. Moreover, in regards to motivational variables, there is a higher correlation between the existence of interpersonal motives for going online and the following two variables: using technology for participative purposes (sharing information, giving opinions, keeping up to date, and training) and the importance of the socioemotional possibilities of a virtual environment (expressing one's own ideas, knowing what others think about one, enjoying a feeling of synchrony or membership). Hence, online environments need to be designed to encourage participative actions by considering identity as a relational process for learning with others and from others [25].

The results showed that the relationships between variables recorded three levels in the expected order: those senior learners with more motivation achieved a higher digital identity level, used the internet for participatory purposes, and considered socioemotional reasons for digital engagement. However, the correlational analyses showed that those who perceived more social support and less loneliness were more motivated to use the internet for information, while seniors who perceived low social support and higher loneliness reported motives for entertainment, the highest digital identity level, and connecting purposes.

Furthermore, the results showed that digital identity levels play a different role depending on the motivations for going online. Only the first level of digital identity, location, has a direct effect when the main reason for using the internet is informative. However, when the reason is entertainment, in addition to location-based digital identity, the significance level also had a direct effect. Finally, when the reason was interpersonal communication, all three factors of digital identity—location, action, and significance—played a relevant role and explained a significant percentage of the variance. The differential effect of digital identity to explain the motivation of internet use is an indicator of construct validity.

The first motive, using the internet to seek out information, seems to have three clear explanatory components: location, a clear purpose based on the reception of information, and the desire for the

content presented in virtual environments to be carefully designed in its formal aspects. When the motive is entertainment, significance is also present as an explanatory factor, along with the desire to overcome perceived loneliness. Finally, when the motive is focused on interpersonal communication, the number of explanatory factors is the highest: the three levels of digital identity are present, the main connecting purpose is based on active participation, the virtual environment's design is geared toward addressing socioemotional aspects, and there is a desire on the part of older adults to overcome their perceived loneliness.

These results reveal the road to be followed to understand education for development as a pedagogic rationale for active aging. Aging cannot ignore social changes, and is not the same as a transitory health crisis. From the perspective of sustainable education, today' techno-cultural society calls for a reappraisal of the formats of social services, lifelong training, and education for citizenship. The results show the different routes for rethinking these arguments. The way of doing so requires starting with an understanding of the digital adult, both potentially and effectively.

Our results supported the conclusion that older adults have generated a digital identity based, mainly, on location variables (convenience, location, familiarity), which is manifested in all of the environments they encounter (connecting purposes), whether their motive for going online is for information, communication, or entertainment. This could be seen as an adaptation to the virtual environment, but not as the specific identification expected of the higher levels of digital identity, such as friendship, membership, sharing, or socializing, or even the significance in the third level, completeness or interdependence. Considering the evolution among the three levels, as explained in the theoretical at framework, this first level is the foundation for further advancement. However, location identity is probably recreated by implicit processes when older adults assume a repertoire of possible movements or (inter)actions. Hence, this adaptive process could be problematic if it leads older adults to accommodate their movements to enclosed actions and environmental requirements, with a possibility of overlooking their own needs, goals, and desires in favor of virtual self-construction. We need to create virtual environments that are likewise scenarios for nurturing personal and social development. This involves favoring the value of personal autonomy and engagement. Sustainable human development informs the need to understand that whatever a person's age, it is important to have the capacity for autonomy and adaptation linked to the value of equality in living conditions, through the acknowledgement of diversity and human dignity, even when one is old. This adaptation would channel the entire process of digital identity reconstruction, running the risk of limiting the ways older adults can be online or develop their digital identities towards further levels.

As technology has predominantly been designed as an organic medium [9]: neutral, impersonal, and able to be managed by the maximum number of users, our results indicate that in the case of active older adults, as a heterogeneous group [20,84], personalization is one of the main indicators for achieving the holistic and sustainable development of this population in our digital societies. The complexity and the articulation of practices on the internet in older adults could make a difference to their digital identity, in view of the correlations between different levels, motives, or uses. We are living through an important time for aging; aging in a sustainable manner means accumulating experiences that involve processes of adjustment to new digital environments and building these environments according to the level of identity in which they are situated. Older adults want to live longer, but they also want those years to be rewarding, alluding to the axiom of the need for integral and sustainable education throughout life.

## 5. Limitations and Future Research

The exclusion of data from large groups of older adults that do not attend formal training reduces the generalization of the findings to a cohort of well-educated participants that are already partaking in the digital society. Further research will need to target a wider range of active older adults, without ignoring the different digital divides [84]. Due to access barriers, a previous level could exist—a zeroth level of identity—in which older adults are frustrated in this initial step of location and are unable

to have a sense of being in the environment, because the very action of entering or leaving presents a challenge, even consider the facing of age-based barriers (vision disabilities, audio impairment, or distractions). More research needs to be done in other countries, as studies have demonstrated the importance of cultural background to digital engagement [85]. Likewise, future researchers need to focus on the development of longitudinal studies to analyze the progression of senior learners' digital identity as an evolving process.

Future researchers also need to focus on the development of longitudinal studies to analyze the progression of senior learners' digital identity as a relational process; this is one of the recommendations made in the 2030 Agenda and the Digital Agenda for Europe 2020, which pays particular attention to older adults, advocating policies on a user-friendly single digital market, whose products and services cater to the specific needs of older adults, fostering digital literacy, the development of skills, and their inclusion. The results stress the pertinence of studying social inclusion according to the premises of education for development, based on these new forms of social inequality linked to different digital profiles for accessing the internet.

Qualitative studies are also needed, seeking stronger clarification of the bridges that enable them to advance to more social online activities, as a prelude to the sociability necessary for active learning, digital inclusion, and sustainable adult education. Even though the results of the factor analysis showed that there are three sub-scales within the overall measure, and they are all significantly correlated with one another; the current research design did not allow presenting evidence for the hierarchical conceptualization of digital identity, or even regressive states; therefore, future studies should be conducted with longitudinal designs to test such conditions.

In addition, experimental studies could confirm the relevance of the socioemotional reasons detected in our study by testing digital learning designs that provide more intense interpersonal communication and participation. Finally, studies should further explore older adults' motives for going online; beyond the three reasons described by Wolfradt and Doll [71], it becomes a requisite to consider active learning as a new motive, because 18.5% of the senior learners in our study revealed that they have occasionally used online training programs. In the near future, older adults could be a growing cohort of virtual learning environment users, who, according to our results, will probably require social connections and knowledge-based interactions for their ongoing inclusion in digital culture. The Sustainable Development Goals (SDG) and the 2030 Agenda will not be achieved without the conscious and intentional inclusion of older adults. A future mainstream study with digitality at its core that identifies best practices with older adults in the implementation of SDGs in specific settings would be a good indicator of the level of social maturity and cultural sustainability that society is achieving. Promoting active aging from the perspective of sustainable development requires us to foster personal and social responsibility throughout one's whole life or entails the creation of virtual environments adapted to older adults' personal and social needs and decisions.

**Supplementary Materials:** The following are available online at http://www.mdpi.com/2071-1050/12/24/10657/s1.

**Author Contributions:** All the authors made substantial contributions to the conception and design of the work, the analysis, and the consideration of the main conclusions. All the authors approved the submitted version. All the authors agreed to be personally accountable for their own contributions and for ensuring that questions related to the accuracy or integrity of any part of the work, even ones in which the author was not personally involved, are appropriately investigated, resolved, and documented in the literature. The individual contributions to the sections of the manuscript are specified as follows: introduction and conclusions: J.M.M.-R. and M.J.H.-S.; material, method and formal analysis: C.T.; writing—original draft preparation, review and editing: J.M.M.-R., M.J.H.-S., and C.T. All authors have read and agreed to the published version of the manuscript.

**Funding:** This research was funded by two projects: a national research projects: "CONNECT-ID. La identidad hiperconectada de la juventud y su percepción del tiempo en el ocio digital" [Hyperconnected identity of young people and their perception of time in digital pastimes] funded by Spain's Ministry of Science, Innovation and Universities (Ref. PGC2018-097884-B-I00); and a regional projects: ""Digital identities in hyperconnected young people: challenges for the family, social, and school context". Regional Government of Castillla y Leon (Ref. SA038G19).

**Conflicts of Interest:** The authors declare no conflict of interest. The funders had no role in the design of the study; in the collection, analyses, or interpretation of data; in the writing of the manuscript; or in the decision to publish the results.

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
