# Peer review of "Digital Identity Levels in Older Learners: A New Focus for Sustainable Lifelong Education and Inclusion"

_sustainability, doi:10.3390/su122410657_

Round 1

Reviewer 1 Report

The subject or the paper is very interesting but not related with the SI. The work is focused on Digital Identity but Sustainable Education is very superficial. In fact, there are almost no references to this field in the paper.

This is why I have marked Must be improved on all items.

The authors should reformulate the work and integrate the Sustainable Education.

Author Response

Dear Reviewer,

Thank you very much for your suggestions to improve the manuscript.

Corrections have been made in many parts of the article in accordance with the reviewers’ comments. The reviewers’ requests for each section and the changes made by the authors are presented below and in the document (track-changes):

Reviewer

Comment/suggestion

Amendments

1

The subject or the paper is very interesting but not related with the SI. The work is focused on Digital Identity but Sustainable Education is very superficial. In fact, there are almost no references to this field in the paper […]

The authors should reformulate the work and integrate the Sustainable Education.

In lines:

Page 2, lines 103-113; 121-132

Page 3, lines 393-397; 409-414

Page 10; Lines 1859-1861; 1862-1872; 1873-1934

Page 11; lines 1935-1943

Page 12; lines 2173-2178; 21912198; 2205-2209.

Page 13; 2295-2297

New references have been inserted: 3,6,7,8,76,77,78,79,80.

2

I would probably move on the "conclusions" to the limitation paragraph.

The authors preferred to separate, since conclusions and discussion links to the results for the sustainability adult education, while limitations and future research links to methodological issues. In any case we can move to same section or separating to subsections in the final version, if it is considered for improvement.

2

Line 561 ends with a comma, whereas it should be a dot.

Corrected.

2 and 3

Probably it could have been interesting to know something more on the formal lifelong learning programs in which the incidental sample was engaged (on a descriptive perspective). 

Detailed information is added in lines 1088-1110.

4

I understand that the theoretical framework includes a zero level (how to help the many older adults who supposedly are in this situation is a different matter) and the section on limitations and future research includes a word of precaution on generalization. But it is a fact that the results are as one would expect from inquiring such a sample and far from innovative.

This consideration is presented, now, in the limitations (lines 2204 to 2209)

About the generalization, an Annex have been included with the validated scale, considering that results from this study could be replicated with the same group of older-adults, or even facilitating the application to other samples, separating into subscales by the three levels.

4

Check if the number of 80% of retired older adults is correct (as 46% have less than 65 or 65, this seems a big number).

Its correct, since one of the reasons for enrolling this kind of programs is having time after retirement, mostly have benefited from early retirement. Also, in this category there are included older adults having a pension, which is now added.

4

There’s an error on table 1 – Age – n(%) – 228 (340,6).

Corrected.

4

Please re-check the language.

The paper has been sent to professional proofreading services and some large sentences are changed along with some wordy expressions.

Reviewer 2 Report

This is a well-conceived and well-written article on adult  digital identity. Both the  theoretical framework (with three different levels of digital identity) and the methodology are well articulated and well explined with much detail. The also  discssion also provides insights into the notion of digital indentity of older adults.  I believe it will an interesting paper for a wide variety of readers. 

Author Response

(The authors gave the same response as above.)

Reviewer 3 Report

I would probably move on the "conclusions" to the limitation paragraph.

LIne 561 ends with a comma, whereas it should be a dot.

Probably it could have been interesting to know something more on the formal lifelong learning programs in which the incidental sample was engaged (on a descriptive perspective). 

Author Response

(The authors gave the same response as above.)

Reviewer 4 Report

This is a well-written and well-structured paper with a high level of internal coherence. The authors are capable of convincing us on the relative importance of the theme, the research and partially of the results.

The theoretical framework is a careful construction, well-sustained and articulated with the methods / results/ conclusions. Of course, the study benefits from an “easy sample”: a set of urban-living older adults who are mostly still working and using the internet for working purposes, are well-educated and plus participating in learning. It is widely recognized (actually, decades ago) that the problems lie elsewhere, for example on how to “bring” to learning or education the ones who are systematically excluded in terms of social structure. I understand that the theoretical framework includes a zero level (how to help the many older adults who supposedly are in this situation is a different matter) and the section on limitations and future research includes a word of precaution on generalization. But it is a fact that the results are as one would expect from inquiring such a sample and far from innovative. This comment, however, is only intended to refute the author’s claims of innovation or originality. The paper is, as I said before, a good, consistent and coherent paper.

Having said this, I think it is a publishable paper with minor corrections. Please re-check the language. Check if the number of 80% of retired older adults is correct (as 46% have less than 65 or 65, this seems a big number). There’s an error on table 1 – Age – n(%) – 228 (340,6). And congratulations.

Author Response

(The authors gave the same response as above.)

Round 2

Reviewer 1 Report

Dear authors,

Thank very mucho for your work.

I think that the paper improved notably.

The link with Education for Sustainability is now visible in the study.

Congratulations.